# CapeX: Category-Agnostic Pose Estimation from Textual Point Explanation

**Matan Rusanovsky, Or Hirschorn and Shai Avidan**
Tel Aviv University
{matanru,orhirschorn}@mail.tau.ac.il and avidan@eng.tau.ac.il

## Abstract

Conventional 2D pose estimation models are constrained by their design to specific object categories. This limits their applicability to predefined objects. To overcome these limitations, category-agnostic pose estimation (CAPE) emerged as a solution. CAPE aims to facilitate keypoint localization for diverse object categories using a unified model, which can generalize from minimal annotated support images. Recent CAPE works have produced object poses based on arbitrary keypoint definitions annotated on a user-provided support image. Our work departs from conventional CAPE methods, which require a support image, by adopting a text-based approach instead of the support image. Specifically, we use a pose-graph, where nodes represent keypoints that are described with text. This representation takes advantage of the abstraction of text descriptions and the structure imposed by the graph. Our approach effectively breaks symmetry, preserves structure, and improves occlusion handling. We validate our novel approach using the MP-100 benchmark, a comprehensive dataset covering over 100 categories and 18,000 images. MP-100 is structured so that the evaluation categories are unseen during training, making it especially suited for CAPE. Under a 1-shot setting, our solution achieves a notable performance boost of 1.26%, establishing a new state-of-the-art for CAPE. Additionally, we enhance the dataset by providing text description annotations for both training and testing. We also include alternative text annotations specifically for testing the model's ability to generalize across different textual descriptions, further increasing its value for future research. Our code and dataset are publicly available at https://github.com/matanr/capex.

## 1 Introduction

Pose estimation deals with the prediction of semantic parts' positions within objects depicted in images, a task crucial for applications like zoology, autonomous driving, virtual reality, and robotics Xu et al. (2022). Previous pose estimation methods were typically constrained by their reliance on category-specific datasets for training. Consequently, when confronted with novel objects, these methods often exhibit limited efficacy due to their lack of adaptability.

To address this challenge, recent research has introduced category-agnostic pose estimation (CAPE) Xu et al. (2022), a paradigm capable of localizing semantic parts across diverse object categories, based on a single or few support examples. To evaluate how category-agnostic a CAPE model is, its pose estimation performance is tested on categories unseen during training. Previous CAPE approaches require a small set of support samples, typically images, annotated with the keypoints of interest. These support images are used to find the best spatial arrangement of the keypoints in the query image, based on latent visual correspondence to the annotated support keypoints.

This raises two challenges. First, the need to provide annotated support image(s) is cumbersome. Second, relying solely on visual correspondence between keypoints in different images, even from the same category, may lead to suboptimal results. This is because no two distinct images share parts with the exact same appearance. Still, both images should share parts with the same semantic meaning. For example, all cats have a head, legs, and a tail, but they never look the same. This idea is even more crucial when the objective is to estimate poses of objects in images from novel categories (i.e., dogs), as in CAPE.

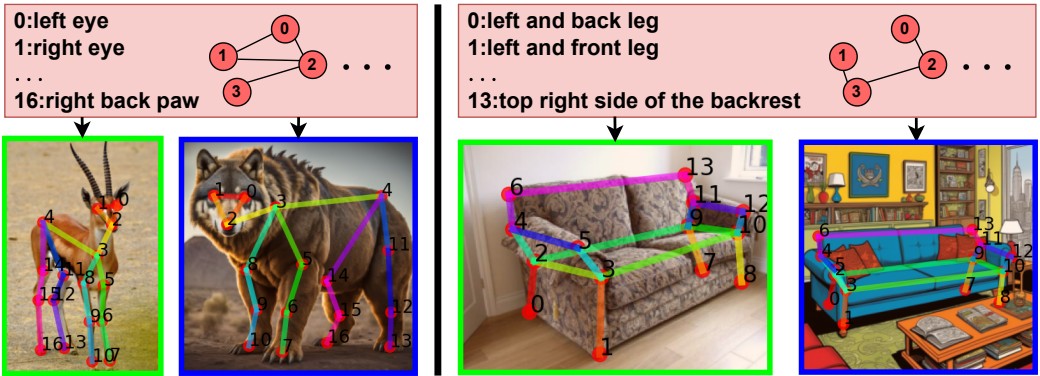

Figure 1: **CapeX in action:** Previous category-agnostic pose estimation (CAPE) methods localize keypoints on a query image using user-provided annotated points from a support image (or images). In contrast, our model localizes the skeleton on query images given support keypoints text descriptions and a corresponding skeleton (in pink). Top is the input support text-graphs, and below each support input, there is a query image from the test set on the left (green), and an AI generated query image on the right (blue). Our approach does not require a support image. Instead, it utilizes the abstraction power of text to improve keypoint localization.

We cope with these limitations by adopting a holistic approach to pose estimation, based on a support graph as input, with textual descriptions on its nodes. No support images are needed. Instead of exclusively relying on visual support data, we leverage the abstraction power of textual data. This comprehensive view enables us to match the query keypoints' appearance to the textual description of the support keypoints, eliminating the need for support images altogether. Furthermore, following GraphCape Hirschorn & Avidan (2023), instead of treating the input keypoints as isolated entities, we treat them as structure-aware connected nodes of a graph. By doing so, we effectively leverage the inherent relationships between keypoints, enhancing the overall performance, breaking symmetry, preserving structure, and better handling occlusions. Figure 1 demonstrates our approach.

To evaluate the efficacy of our proposed method, we utilize the extended version Hirschorn & Avidan (2023) of the CAPE benchmark, MP-100 Xu et al. (2022). This dataset consists of more than 18,000 images spanning 100 categories, encompassing diverse subjects such as animals, vehicles, furniture, and clothes. Importantly, MP-100 is structured so that the test categories are entirely unseen during training and validation, making it particularly suited for CAPE. As some of the categories miss the keypoints' text descriptions, we collected and unified the text descriptions of the keypoints in all categories. Our method is the first text-based CAPE method that was evaluated on the full MP-100 dataset using only textual support information, against previous CAPE methodologies. Notably, our approach surpasses the performance of existing CAPE methods, showcasing a new state-of-the-art performance under the 1-shot setting.

In summary, our contributions can be outlined as follows:

- We propose modeling the support keypoints using connected graph nodes coupled with text descriptions as opposed to previous methods that rely on visual signals, or independent textual signals. This methodology matches the support to the query keypoints, thanks to the abstraction power of text and graphs. Furthermore, this approach does not require support images for either training or inference.

- We provide an enhanced version of the MP-100 dataset with textual annotations for the keypoints in all categories, enriching the benchmarking capabilities for category-agnostic pose estimation. We also include alternative annotations to test the ability of the framework to adapt to text modifications in inference.

- We extensively evaluate our method against other CAPE approaches and text-based animal pose estimation techniques. Our method is the first text-prompt-based technique evaluated as a CAPE method on the full MP-100 dataset. We set a new benchmark in category-agnostic pose estimation, achieving state-of-the-art performance on MP-100 without fine-tuning the support feature extraction process.

## 2 RELATED WORK

Recent CAPE approaches are either image-based and offer improvements to the previous methods, while a new branch of research explore utilizing text-based prompts for pose estimation. However, prior to this work, text-based pose estimation approaches were not tested on the full MP-100 benchmark for CAPE.

### 2.1 IMAGE-BASED CATEGORY-AGNOSTIC POSE ESTIMATION

The primary aim of pose estimation is to localize the semantic keypoints of objects or instances precisely. Traditionally, pose estimation methods have been largely tailored to specific categories, such as humans Fang et al. (2022); Cao et al. (2019); Yang et al. (2021), animals Yu et al. (2021); Yang et al. (2022), or vehicles Song et al. (2019); Reddy et al. (2018). However, these prior works are constrained to object categories encountered during training.

An emerging aspect in this field is category-independent pose estimation, as introduced by POM-Net Xu et al. (2022). This few-shot approach predicts keypoints by comparing support keypoints with query images in the embedding space, addressing the challenge of object categories not seen during training. POMNet employs a transformer to encode the support keypoints and query images. It uses a regression head to predict similarity from the extracted features. CapeFormer Shi et al. (2023) extends this matching paradigm to a two-stage framework, refining unreliable matching outcomes to improve prediction precision. A multitude of recent works suggested that it is beneficial to not only use the local support features as a goal for query keypoints mining, but to also explore the global semantics and structure hidden in the support data. ESCAPE Nguyen et al. (2024) proposed learning a mixture of super-keypoints representing a prior over the features of the keypoints. When adapting to novel categories, it matches the novel keypoints to the most related super-keypoint, then transfers the encoded knowledge from the matched super-keypoint to the novel keypoints. Meta-Point Chen et al. (2024) indicated that there are some inherent or universal points in each object, that can be identified without any support image. Following this insight the authors suggested maintaining learnable embeddings to capture inherent information of various keypoints, which predict meta-points without the support data. These predicted meta-points are assigned, and then refined according to the support keypoints. SCAPE Liang et al. (2025) proposed a simple yet powerful architecture that integrates global information while accounting for the relationships between keypoints, thereby facilitating mutual assistance in point prediction. GraphCape Hirschorn & Avidan (2023) presented a significant departure from previous CAPE methods, which refer to keypoints as isolated entities, by treating the input pose data as a graph. It utilizes Graph Convolutional Networks (GCNs) to leverage the inherent object's structure to break symmetry, preserve the structure, and better handle occlusions. SDPNet Ren et al. (2024) suggested dynamically predicting category-specific semantic and structural information from the support samples. Then, this information is leveraged for dynamically modulating the interaction between the query points features and query image features.

However, relying solely on visual features presents a challenge in matching semantically similar entities. This is because no two distinct images contain parts that look exactly the same, even though both images may have parts with the same semantic meaning (please see Section 3.1 for further details). This insight suggested that matching via textual-prompts might be beneficial for CAPE. Our work builds upon GraphCape, utilizing its structure-aware architecture, while introducing the abstraction power of text.

### 2.2 TEXT-BASED CATEGORY-AGNOSTIC POSE ESTIMATION

Recently, CLAMP Zhang et al. (2023) leveraged CLIP to prompt animal keypoints. They found that establishing effective connections between pre-trained language models and visual animal keypoints is challenging due to the substantial disparity between text-based descriptions and keypoint visual features. CLAMP attempts to narrow this gap by using contrastive learning to align the text prompts with the animal keypoints during training. CLAMP has demonstrated strong results in animal keypoint detection. However, its contrastive-based design limits its effectiveness with novel text descriptions, a crucial capability in a CAPE setting.

A growing area in computer vision called Open-Vocabulary learning is being explored in various vision tasks. These new methods aim to localize and recognize categories beyond the textual labeled space. The open-vocabulary approach is broader, more practical, and more efficient compared to weakly supervised setups Wu et al. (2024). KDSM Zhang et al. (2024), introduces a framework aimed at addressing the open-vocabulary keypoint detection task. However, KDSM was designed to estimate animal poses and the authors report results on only a subset of the full MP-100 dataset. X-Pose Yang et al. (2023) is a multi-modal framework capable of estimating poses based on both textual and visual prompts. Trained as an end-to-end framework, X-Pose can also detect keypoints of multiple object in a single pass. PPM Peng et al. (2024) proposes learning a pseudo prompt via Stable Diffusion, using a cross-attention map linked to the support image, to approximate the target support keypoint heatmap. These pseudo prompts are then employed to localize corresponding keypoints in the query image. While text-based, this method still requires annotated support images and only implicitly uses the extracted text from them.

Notably, all previous text-prompt-based approaches have demonstrated performance only on textual descriptions that were restricted to animal species. However, X-Pose has also reported visual-prompt-based pose estimation on general categories, and PPM demonstrated implicit use of text out of visual support information. Our approach aims for general keypoint estimation of any category while taking advantage of structure as a prior for localization by treating the input prompts as a text-graph.

## 3  METHOD

### 3.1  DISADVANTAGES OF VISUAL PROMPTS IN CAPE

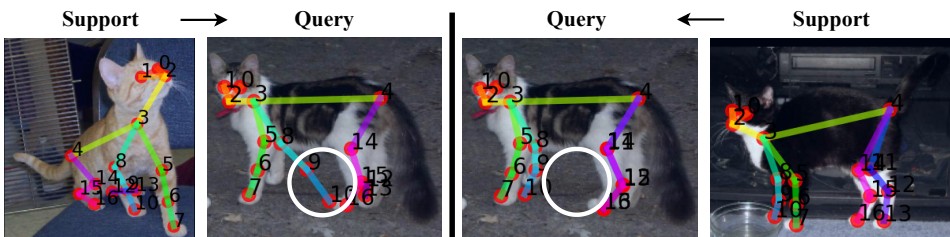

Figure 2: **Visual Prompts Inconsistencies:** Using visual features impairs the ability to describe abstract semantic parts: different support images (left and right images) result in different estimated poses on the *same* query image (center two images). Keypoints definitions and skeletons are the same. Main differences are marked with the white circle. Results were generated using GraphCape.

We exemplify the key disadvantage of support image-based CAPE approaches in Figure 2. Specifically, GraphCape, suffers from incorrect pose estimations when prompted with visually inconsistent support images.

The core idea of our work is that for the task of CAPE, it is more beneficial to describe the searched points in the query image using text description instead of relying only on the visual features of the support images. This is because text allows a higher level of abstraction and offers a looser restriction to the support request. This is true even when the support and query images are from the same category - for example, no two cats share visually the exact same

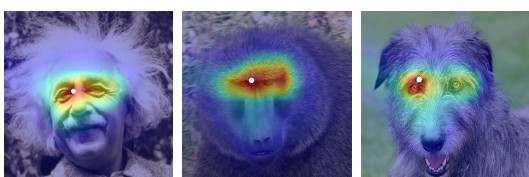

Figure 3: **Text-based cross-attention maps:** When prompted with the same support text: 'top right side of the left eye', our model localizes and attends to the requested abstract semantic concept on different categories of query images. Highest attention values are marked by white circles.

*right front paw*, but both cats have a part within them that follows the same text description: *right front paw*. This distinction is even more significant when dealing with images from different categories as in CAPE.

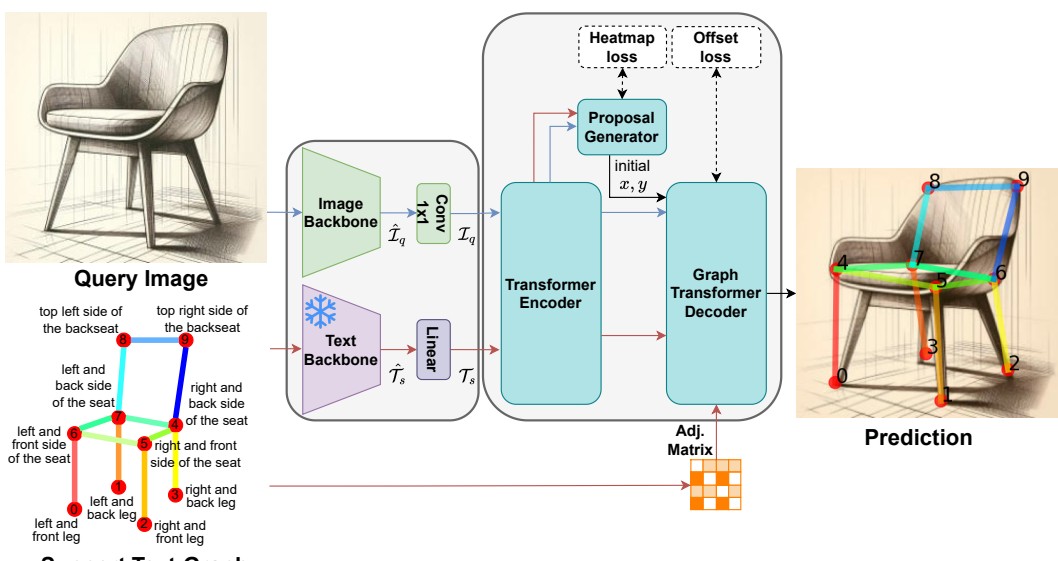

Figure 4: **Architecture overview:** Our framework uses image and text backbones benefiting from both accurate and abstract descriptions respectively. The extracted feature descriptors are forwarded into the transformer encoder that refines them. The refined features are passed into the proposal generator alongside the graph transformer decoder, utilizing the graph structure within the data.

In contrast to image-based methods, we suggest modeling the support information as text, representing semantic abstract parts. In Figure 3, we visualize cross-attention maps that were generated from a text-based model (further details are in Section 3.2). We demonstrate the effectiveness of the same support text description in symmetry breaking and localization of the same semantic concept, on query images from different categories.

## 3.2 TEXT PROMPTS AS SUPPORT DATA

Our framework extracts visual features from the query image and matches them to the textual features that are extracted from the support text-graph. We incorporated this notion by introducing text comprehension into GraphCape's framework Hirschorn & Avidan (2023).

A pre-trained and fine-tuned SwinV2-T Liu et al. (2022) is utilized for extracting image features from the input query image producing the feature map $\hat{\mathcal{I}}_q \in \mathbb{R}^{hw \times C_i}$, where $hw$ is the total number of patches and $C_i$ is the image embedding dimension. Then $\hat{\mathcal{I}}_q$ is passed through a 1x1 convolutional layer, resulting in $\mathcal{I}_q \in \mathbb{R}^{hw \times C}$.

The support keypoint text descriptions are embedded in our model using a pre-trained gte-base-v1.5 Li et al. (2023). The text embeddings of all $K_s$ keypoints of the provided support sample are then normalized. The normalized keypoints are padded with zeros, effectively resulting in $K$ keypoints, where $K$ is defined to be the maximum amount of possible keypoints in the dataset. The final text feature map is of the form $\hat{\mathcal{T}}_s \in \mathbb{R}^{K \times C_t}$ where $C_t$ is the text embedding dimension. Then $\hat{\mathcal{T}}_s$ is passed through a linear layer resulting in $\mathcal{T}_s \in \mathbb{R}^{K \times C}$. During training, the text backbone is frozen. This approach also offers a lighter optimization procedure, as the gradients of the text features are ignored. An architecture overview is presented in Figure 4.

The extracted query image features and the support descriptions features are then refined using the transformer encoder. This encoder comprises three transformer blocks. Since the embedding spaces of the support text and query image differ, the support and query features are first fused together and then separated again. This practice aids in closing the gap between their representations Shi et al. (2023) using self-attention layers. Then, similarity heatmaps between the query and support features are formed, using the proposal generator. The proposal generator utilizes a trainable inner-product mechanism Shi et al. (2022) to explicitly represent similarity. Peaks are then chosen from these maps

Table 1: **MP-100 results:** $PCK_{0.1}$ and $PCK_{0.2}$ performance under the 1-shot setting. † means that the we adapted the architecture to work on MP-100. Our approach outperforms all other methods on average.

| | Model | Support | Split 1 | Split 2 | Split 3 | Split 4 | Split 5 | Avg |
|---|---|---|---|---|---|---|---|---|
| $PCK_{0.1}$ | CapeFormer-T Hirschorn & Avidan (2023) | Image | 74.06 | 70.18 | 67.94 | 68.91 | 71.02 | 70.42 |
| | Pose Anything-T Hirschorn & Avidan (2023) | Image+Graph | 76.45 | 72.21 | 68.42 | 70.65 | 71.40 | 71.83 |
| | CapeX-no-graph | Text | 77.89 | 71.66 | **70.55** | 72.17 | 75.65 | 73.58 |
| | **CapeX** | Text+Graph | **79.52** | **73.60** | 69.68 | **72.97** | **76.13** | **74.38** |
| $PCK_{0.2}$ | ProtoNet Snell et al. (2017) | Image | 46.05 | 40.84 | 49.13 | 43.34 | 44.54 | 44.78 |
| | MAML Finn et al. (2017) | Image | 68.14 | 54.72 | 64.19 | 63.24 | 57.20 | 61.50 |
| | Fine-tuned Nakamura & Harada (2019) | Image | 70.60 | 57.04 | 66.06 | 65.00 | 59.20 | 63.58 |
| | POMNet Xu et al. (2022) | Image | 84.23 | 78.25 | 78.17 | 78.68 | 79.17 | 79.70 |
| | CapeFormer Shi et al. (2023) | Image | 89.45 | 84.88 | 83.59 | 83.53 | 85.09 | 85.31 |
| | CLAMP† Zhang et al. (2023) | Text | 72.37 | | | | | |
| | ESCAPE Nguyen et al. (2024) | Image | 86.89 | 82.55 | 81.25 | 81.72 | 81.32 | 82.74 |
| | X-Pose Yang et al. (2023) | Image\Text | 89.07 | 85.05 | 85.26 | 85.52 | 85.79 | 86.14 |
| | MetaPoint+ Chen et al. (2024) | Image | 90.43 | 85.59 | 84.52 | 84.34 | 85.96 | 86.17 |
| | CapeFormer-T Hirschorn & Avidan (2023) | Image | 89.48 | 86.69 | 85.31 | 84.79 | 84.97 | 86.25 |
| | GraphCape Hirschorn & Avidan (2023) | Image+Graph | 91.19 | 87.81 | 85.68 | 85.87 | 85.61 | 87.23 |
| | SDPNet(HRNet-32) Ren et al. (2024) | Image | 91.54 | 86.72 | 85.49 | 85.77 | 87.26 | 87.36 |
| | PPM+CPT Peng et al. (2024) | Image (+Text) | 91.03 | 88.06 | 84.48 | 86.73 | 87.40 | 87.54 |
| | SCAPE Liang et al. (2025) | Image | 91.67 | 86.87 | **87.29** | 85.01 | 86.92 | 87.55 |
| | CapeX-no-graph | Text | 91.98 | 88.51 | 84.94 | 87.22 | **89.87** | 88.50 |
| | **CapeX** | Text+Graph | **92.79** | **89.47** | 84.95 | **87.25** | 89.61 | **88.81** |

to act as the basis for similarity-aware proposals. A graph transformer decoder network receives these initial proposals, processes them using a combination of attention and Graph Convolutional Network (GCN) layers, and predicts the final estimated keypoints locations. Utilizing GCN layers allows for the explicit consideration of semantic connections between keypoints, thereby benefiting CAPE tasks. Importantly, following GraphCape, we ignore the support keypoint identifiers and do not add their positional encoding before passing them into the transformer encoder. This practice improves performance when tested on keypoints with a random ordering (which is reasonable to support in CAPE settings), but decreases performance when tested on the same expected orderings. Instead, we learn the spatial relations between keypoints via the learned graph structure. Please refer to Hirschorn & Avidan (2023) for further details. We visualize the last cross-attention maps that were generated from our decoder in Figure 3.

To train our end-to-end method we use two loss terms: $\mathcal{L}_{heatmap}$ and $\mathcal{L}_{offset}$. The former penalizes the similarity metric while the latter penalizes the localization output:

$$\mathcal{L}_{heatmap} = \frac{1}{(K \cdot H \cdot W)} \sum_{i=1}^{K} ||\sigma(M_i) - H_i|| \tag{1}$$

$$\mathcal{L}_{offset} = \frac{1}{L} \sum_{i=1}^{L} \sum_{i=1}^{K} |P_i^l - \hat{P}_i| \tag{2}$$

where $\sigma$ is the sigmoid function, and for each point $i$, $M_i$ is the output similarity heatmap of the proposal generator, $H_i$ is the ground truth heatmap, $P_i^l$ is the output location from layer $l$ and $\hat{P}_i$ is the ground truth location. The overall loss is:

$$\mathcal{L} = \lambda_{heatmap} \cdot \mathcal{L}_{heatmap} + \mathcal{L}_{offset} \tag{3}$$

## 4 EXPERIMENTS

In line with prior CAPE studies, we utilize the MP-100 dataset Xu et al. (2022) as both our training and evaluation dataset, which comprises samples sourced from existing category-specific pose estimation datasets Lin et al. (2014); Sagonas et al. (2016); Koestinger et al. (2011); Wang et al. (2018); Ge et al. (2019); Yu et al. (2021); Labuguen et al. (2021); Pereira et al. (2019); Graving et al. (2019); Welinder et al. (2010); Reddy et al. (2018); Khan et al. (2020); Wu et al. (2016). This dataset consists of over 18K images spread across 100 distinct sub-categories and 8 super-categories

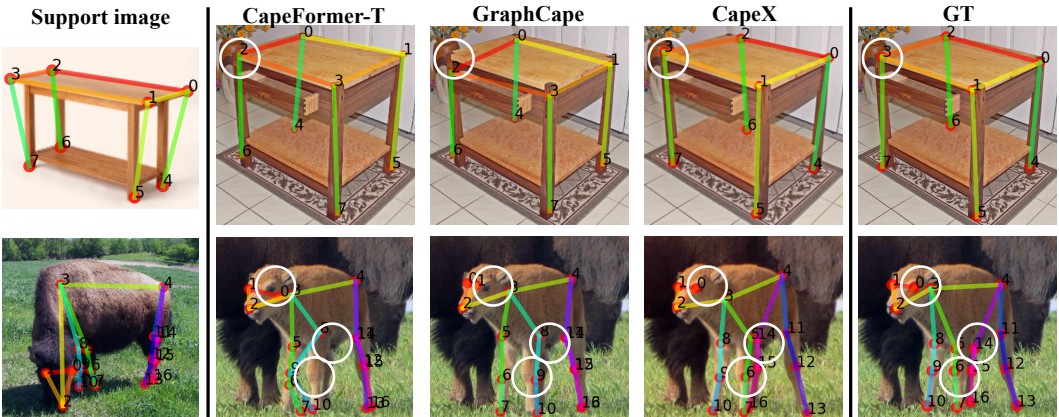

Figure 5: **Qualitative results:** From left to right: support images that are used by the competitors, CapeFormer-T, GraphCape, our model, and the GT. Support text descriptions used by our model are not shown. Main differences are pointed out using white circles. In both examples, CapeFormer-T and GraphCape fail to correctly distinguish between left and right keypoints.

(human hand & face & body, animal face & body, clothes, furniture and vehicle), featuring varying numbers of keypoints, ranging from 8 to 68 keypoints.

The dataset is divided into five separate splits for training and evaluation. Importantly, each split ensures that the categories used for training, validation, and testing are mutually exclusive, ensuring that the categories used for evaluation are unseen during the training phase.

The original dataset comes with partial skeleton annotations in different formats, including variations in the keypoint indexing. We use the updated version of GraphCape Hirschorn & Avidan (2023) that includes unified skeleton definitions for all categories. The updated version predominantly featured brief text sentences describing each point within most categories. However, certain categories exhibited text descriptions with distinct characteristics, such as the use of underscores between words instead of spaces, while others lacked any text descriptions altogether. We annotated and standardized the text descriptions of all points in all categories, offering a new supervision capability to the updated version of Hirschorn & Avidan (2023) of the original MP-100.

To evaluate our proposed architecture for pose estimation across arbitrary categories based on text, we adapted CLAMP Zhang et al. (2023) to operate in a CAPE setting using our text annotations, and compared its performance. CLAMP was originally designed to estimate animal poses using only 17 keypoint descriptions shared across all animals. To extend CLAMP to handle a varying set of keypoint definitions, we increased the number of keypoints to 100 and used masks to focus solely on the visible keypoints. Additionally, we modified the attention layers to prevent attention from being applied to invisible keypoints. In addition, Zhang et al. (2024) introduced a two-setting dataset, MP-78, which is a subset of MP-100, restricted to animals, accompanied with textual annotations to keypoints. Setting A uses a closed set of keypoint descriptions, splitting keypoints into seen and unseen categories. This suits animal-focused works but is less relevant to CAPE. Setting B, analogous to MP-100 and more relevant to CAPE, splits training and testing by seen and unseen animal species and was tested on multiple works.

To assess our model's performance, with and without graphs, we employ the Probability of Correct Keypoint (PCK) metric Yang & Ramanan (2012), setting a PCK threshold of 0.2, following the conventions established by previous works. We also report our model's results with a PCK threshold set to 0.1 and compare them to our baseline models, GraphCape Hirschorn & Avidan (2023) and the enhanced version of CapeFormer Shi et al. (2023): CapeFormer-T Hirschorn & Avidan (2023). More design choices and evaluations are in the supplementary.

**Implementation Details** To ensure a fair comparison, except for the text backbone, the configuration settings remain consistent with GraphCape and CapeFormer. The trainable features of the framework remain exactly the same as in GraphCape (except for the new linear layer) since the text

Table 2: **MP-78 results:** $PCK_{0.2}$ performance under setting B Zhang et al. (2024). Our approach outperforms KDSM Zhang et al. (2024) by a large margin.

Table 3: **Text Modifications:** $PCK_{0.2}$ performance of CapeX trained using the original text descriptions and tested with the modified ones.

| Model | Split 1 | Split 2 | Split 3 | Split 4 | Split 5 | Avg |
|---|---|---|---|---|---|---|
| KDSM | 85.48 | 89.45 | 84.29 | 86.25 | 81.17 | 85.33 |
| **CapeX** | **92.65** | **94.69** | **89.77** | **93.88** | **89.88** | **92.17** |

| Test | Split 1 | Split 2 | Split 3 | Split 4 | Split 5 | Avg |
|---|---|---|---|---|---|---|
| Typo | 79.41 | 81.27 | 73.05 | 74.93 | 78.66 | 77.46 |
| Synonym | 85.93 | 83.16 | 78.61 | 80.29 | 81.55 | 81.91 |
| Translate | 92.43 | 89.06 | 83.81 | 86.62 | 87.59 | 87.90 |
| CapeX | 92.79 | 89.47 | 84.95 | 87.25 | 89.61 | 88.81 |

backbone is frozen in our framework. However, we also evaluate and present the performance of the framework with an unfrozen text backbone in the supplemental Table 5. $C_i$ is 768 in SwinV2-T, $C_t$ is 768 in gte-base-v1.5. $C$ and $K$ are set to 256 and 100, respectively. The architecture is implemented within the MMPose framework Contributors (2020), trained using the Adam optimizer for 200 epochs with a batch size of 16. The initial learning rate is $10^{-5}$, reducing by a factor of 10 at the 160th and 180th epochs. Our model requires 6.5 GB of GPU memory and takes roughly 13 hours to train for each split, on a machine equipped with an NVIDIA RTX A5000 GPU.

## 4.1 BENCHMARK RESULTS

We conduct a comparative analysis of our approach with gte-base-v1.5 Li et al. (2023) as the freezed text backbone, against GraphCape Hirschorn & Avidan (2023) as our baseline, PPM+CPT Peng et al. (2024), SDPNet Ren et al. (2024), ESCAPE Nguyen et al. (2024), MetaPoint Chen et al. (2024), X-Pose Yang et al. (2023), SCAPE Liang et al. (2025), our adaption of CLAMP Zhang et al. (2023) to CAPE, as well as prior CAPE methodologies such as CapeFormer Shi et al. (2023) and its enhanced version CapeFormer-T from Hirschorn & Avidan (2023), POMNet Xu et al. (2022), ProtoNet Snell et al. (2017), MAML Finn et al. (2017), and Fine-tuned Nakamura & Harada (2019).

Our evaluation is based on the MP-100 dataset, considering the 1-shot scenario. While traditionally 1-shot refers to a single required support image, our framework uses a single text-graph instead. We do not report the 5-shot results, since all 5 support samples share the same text-graph and therefore do not introduce any new information compared to the 1-shot setting. The results are presented in Table 1. Not surprisingly, CLAMP performed poorly on general categories with different textual annotations, because of its contrastive-learning nature. Since CLAMP showed non-competitive results in the CAPE setting, we report its results only for split 1. We find that utilizing the graph structure via the graph transformer decoder as in Hirschorn & Avidan (2023) slightly boosts the performance compared to the the original MLP based transformer decoder as in Shi et al. (2023) (+0.31%). Our graph-less version offers a margin of 3.19% on average in comparison to our image-based baseline that does not utilize graphs, CapeFormer-T. Notably, our text-based approach outperforms our image-based baseline, GraphCape on most splits, with an average improvement of 1.58%. Interestingly, the performance gap between our model and GraphCape gets bigger when $PCK_{0.1}$ is compared: 2.55%.

In addition, our model surpasses the previous state-of-the-art SCAPE with a margin of 1.26%. These results establish a new state-of-the-art result, showcasing the efficacy of utilizing text-graphs for CAPE. We also compare our work to KDSM Zhang et al. (2024) using setting B in MP-78, employing the same text annotations as KDSM. Results are shown in Table 2, where our model outperforms KDSM by a large margin.

Figure 5 presents a qualitative comparison between our model, the visual-prompt-based baseline that uses graphs (GraphCape), and the visual-prompt-based baseline without graphs (CapeFormer-T). Our model performs well given the support text-graph input (not shown), while the support image-based techniques are sensitive to the inconsistencies between the support and query images.

## 4.2 ABLATION STUDY

We now present key ablation experiments. Additional ablations (including alternative architectures, occlusion handling, and additional examples) can be found in Section A.3.

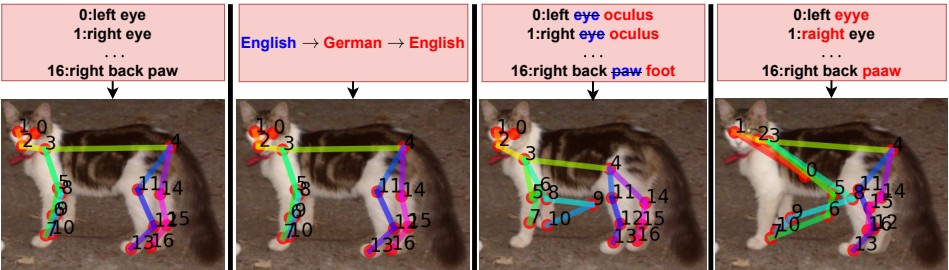

Figure 6: **Modified text descriptions:** Top is the support keypoints descriptions. Left is a prediction with the **original** text annotation, followed by **translation** test annotations, **synonym** test annotations, and **typo** test annotations, respectively. Below each description, query output(s) are presented.

**Text Modifications:** We test the robustness of our model on different types of modifications for keypoint descriptions in Table 3, and show a qualitative example in Figure 6. Specifically, we test the model's adaptability to translations into another language and back to English, to synonym substitutions, and to typos. These tests assess the model's ability to handle text modifications across varying levels of difficulty: translations often remain close to the original descriptions, synonyms preserve the same semantic meaning, and typos can sometimes disrupt the original content. The levels of difficulty are reflected in the results presented in Table 3 and Figure 6. The text backbone's lack of fine-tuning during training is a key factor in its robustness, as it prevents overfitting to text descriptions from the training set. As a result, the model handles modified text inputs effectively while maintaining consistent pose estimations. This testing approach relies on alternative annotations introduced during evaluation, offering a benchmark for assessing text-based CAPE models. We encourage future text-based CAPE studies to assess their performance on these test annotations to advance towards open-vocabulary capabilities.

**Different Graphs:** Unlike image-based CAPE approaches which rely on spatial locations within an image, our approach uses a list of textual descriptions representing keypoints as input. This allows us to query a large language model to construct a skeleton tailored to the provided texts. We evaluated our model, trained on split 1 with the original graphs, using the test set where the connectivity information for each category was replaced with custom edges generated by ChatGPT OpenAI (2024), based solely on the textual descriptions. Additionally, we test the performance using fully connected as well as empty graphs. The results are summarized in Table 4. The findings are twofold. First, the results prove that our model utilizes the graphs' information since meaningful graph structures are superior to fully-connected or empty graphs. Second, the results demonstrate

Table 4: **Different Graphs:** $PCK_{0.2}$ performance of CapeX trained using the original graphs and tested with the different ones.

| Provided Graphs | Split 1 |
|---|---|
| Fully-Connected Graphs | 90.23 |
| Empty Graphs (no edges) | 91.27 |
| LLM-created Graphs | 91.76 |
| CapeX | 92.79 |

the generalizability of our model. When provided with an automatically generated graph during inference, the model trained on manually annotated graphs produced results that outperformed those using fully connected or empty graph structures. See Section A.2.2 for further details.

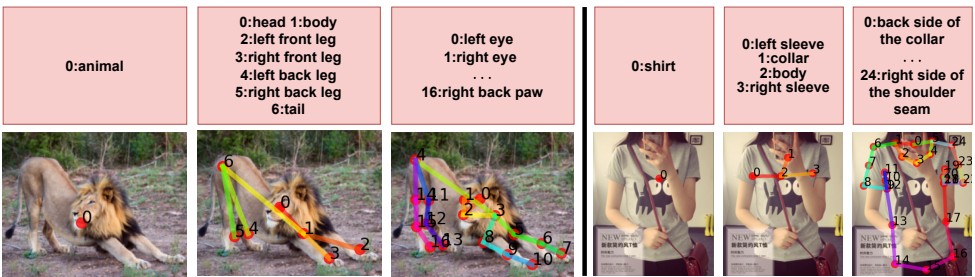

Figure 7: **Text Abstractions:** Model performance over different levels of text-pose abstractions.

**Abstraction:** We test the robustness of our model on keypoints described using varying levels of textual and pose-graph abstractions —— a task that is challenging when relying only on support visual cues. Results are in Figure 7. Although not trained with the prompted text descriptions and pose-graphs, the model presents satisfactory results in both examples.

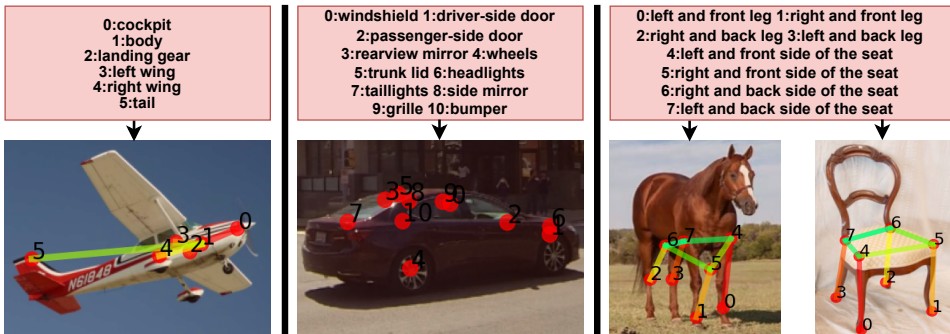

Figure 8: **Failure cases:** From left to right: a category outside of the dataset, introducing vastly new keypoint descriptions, and cross-category descriptions.

## 5 Limitations

**Failure Cases:** We stress that although our model strives for full open-vocabulary performance, it is still trained on a relatively small training set over arguably a short time period, compared to the state-of-the-art large vision-language foundation models. We present in Figure 8 a few failure cases that may be addressed in future research. Our model does not handle new categories with novel text-graphs well, as can be seen in the plane example on the left (left and right wings). In addition, prompting with vastly new parts may lead to incorrect localizations as can be seen in the car example on the middle (for example, grille or bumper). On the right panel, the model incorrectly executes semantically challenging descriptions. For example, the model can not localize a 'seat' in a horse, even though riders may sit on it. Instead, it hallucinates a pose of a chair that it has seen in the training set. Additionally, we conducted a cross super-category ablation to assess our model's ability to generalize to entirely new categories (see supplemental Table 7). This revealed a limitation: the model struggles with symmetry when no similar examples are available. Unlike image-based methods, which leverage spatial context (e.g., identifying a table's front left leg from a support keypoint), our approach relies on textual labels, where terms like "left" or "front" can be interpreted inconsistently. See Section A.3.2 for details.

**Text or Graphs as Support:** Lastly, we note that a key limitation of using text for arbitrarily annotated keypoints is the semantic similarity among descriptions of similar keypoints. While this can challenge text-based CAPE approaches, our text-graph method overcomes it by learning relationships among keypoints with similar or identical names. For example, it successfully matches multiple "left sleeve" keypoints, as shown in Figure 7. Furthermore, although our framework requires complementary graph annotations, Table 4 demonstrates its effectiveness even with graphs automatically generated from text annotations using a large language model. Section A.2 extends the discussion on the limitations of using text or graphs, and on the benefits in combining them as support information for CAPE.

## 6 Conclusions

CapeX is a CAPE approach that takes a pose-graph, where text descriptions are attached to its nodes, and finds these keypoints in a query image. This stands in contrast to previous CAPE approaches that require support image with annotated pose-graph as part of the input, or support texts without any structural connectivity information. CapeX was tested on the standard MP-100 and on the animals-restricted MP-78 datasets and achieves new state of the art results, surpassing previous CAPE methods that rely on support images as well as support texts.

ACKNOWLEDGMENTS

This research was partially funded by ISF grant 2132/23.

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

# A    APPENDIX / SUPPLEMENTAL MATERIAL

## A.1    ETHICAL STATEMENT

Advancements in pose estimation technology can revolutionize fields such as autonomous driving, smart cities, sports, etc., by enabling precise movement analysis. However, when a pose estimation tool is used, specifically in fields such as surveillance, it is crucial to address privacy concerns and establish ethical guidelines to protect sensitive personal data and ensure responsible use.

## A.2    LIMITATIONS OF SUPPORT INFORMATION FOR CAPE

### A.2.1    TEXT AS A SUPPORT INFORMATION FOR CAPE

A key limitation of using text as support information for arbitrarily annotated keypoints is the semantic similarity among descriptions of similar keypoints. For animal categories, the distinctions between keypoints are generally clear, as demonstrated by the following example:

['left eye', 'right eye', 'nose', 'neck', 'root of tail', 'left shoulder', 'left elbow', 'left front paw', 'right shoulder', 'right elbow', 'right front paw', 'left hip', 'left knee', 'left back paw', 'right hip', 'right knee', 'right back paw']

However, for other categories, such as short sleeved shirt, these distinctions can be far less clear, with multiple keypoints sharing similar descriptions, such as right/left sleeve:

['back side of the collar', 'left side of the collar', 'front left side of the collar', 'front side of the collar', 'front right side of the collar', 'right side of the collar', 'left side of the shoulder seam', 'left sleeve', 'left sleeve', 'left sleeve', 'left sleeve', 'left sleeve', 'left and upper body of the shirt', 'left and lower body of the shirt', 'bottom left side of the shirt', 'bottom side of the shirt', 'bottom right side of the shirt', 'right and lower body of the shirt', 'right and upper body of the shirt', 'right sleeve', 'right sleeve', 'right sleeve', 'right sleeve', 'right sleeve', 'right side of the shoulder seam']

The complete set of added annotations, including the 68 keypoints for the human face (which also exhibit extensive repetition), is provided in the supplemental material.

While these repetitive annotations may challenge text-based CAPE approaches, our text-graph method addresses this issue by effectively learning relationships among keypoints with similar or identical names. For instance, our approach can relate:

['left side of the shoulder seam', 'left sleeve', 'left sleeve', 'left sleeve', 'left sleeve', 'left sleeve', 'left and upper body of the shirt']

and appropriately match them to the query image, as demonstrated in the right panel of Figure 7.

Additionally, we evaluate our model using custom skeleton connections. Specifically, training and testing with an empty graph configuration (representing text-only support) yields an average PCK0.2 of 88.50, compared to 88.81 with the original skeleton (Table 1). These results underscore the importance of graph structures in effectively managing repetitive annotations and improving performance.

### A.2.2    GRAPHS AS A SUPPORT INFORMATION FOR CAPE

Although our framework requires complementary graph annotations, graph annotations are required per category, not per instance. In fact, for the whole MP-100 dataset, the authors of Graph-Cape Hirschorn & Avidan (2023) were required to manually annotate only 22 missing skeleton definitions, since connectivity data was available for most categories.

Furthermore, unlike previous image-based CAPE approaches, our text-graph approach allows for the easy integration of automatically generated graph structures. Since our support input consists of a list of texts representing keypoints (in comparison to locations within an image), we can simply query an LLM model to construct a skeleton suited for the provided texts. We tested how our model that was trained on split 1 with the original graphs performed on the test set where the connectivity information for each category was replaced with the custom edges that were produced by ChatGPT OpenAI (2024) (provided with only the texts), as well as with a fully connected skeleton, and with an empty graph. The results are in Table 4.

We attribute the decrease in performance of the fully connected graph compared to the empty graph to the fact that the model takes unnecessary relationships between keypoints under consideration. For example, when a connection between the left eye and the right back paw of a horse is provided, the model unnecessarily predicts the position of the left eye based also on the right back paw (and vice versa). When an empty graph is provided as input this problem is mitigated.

The results highlight the generalization capacity of our model, since when an automatically-generated graph was used on inference (on a model that was trained on manually-annotated graphs), the results were superior compared to fully-connected or empty graphs.

## A.3 ADDITIONAL EXPERIMENTS

### A.3.1 DIFFERENT ARCHITECTURES

Table 5: **Ablation experiments of CapeX:** Tuning VS. Freezing the text backbone in model training, utilizing the graph transformer decoder or the original MLP transformer decoder. $PCK_{0.2}$ performance under 1-shot setting, with gte-base-v1.5 or CLIP ViT-B/32 as the text backbone and SwinV2-T or SwinV2-S as the image backbone. Our method is marked in bold.

| Image Backbone | Text Backbone | | Decoder Type | Split 1 | Split 2 | Split 3 | Split 4 | Split 5 | Avg |
|---|---|---|---|---|---|---|---|---|---|
| SwinV2-T | CLIP ViT-B/32 | Freezed | Graph | 92.63 | 88.01 | 84.61 | 87.45 | 89.50 | 88.44 |
| SwinV2-T | gte-base-v1.5 | Tuned | Graph | 93.09 | 87.77 | 85.12 | 87.50 | 88.05 | 88.31 |
| SwinV2-T | gte-base-v1.5 | Freezed | MLP | 91.98 | 88.51 | 84.94 | 87.22 | 89.87 | 88.50 |
| **SwinV2-T** | **gte-base-v1.5** | **Freezed** | **Graph** | **92.79** | **89.47** | **84.95** | **87.25** | **89.61** | **88.81** |
| SwinV2-S | CLIP ViT-B/32 | Freezed | Graph | 95.17 | 88.88 | 87.72 | 88.24 | 91.81 | 90.37 |
| SwinV2-S | gte-base-v1.5 | Tuned | Graph | 96.28 | 89.15 | 89.17 | 87.66 | 92.62 | 90.98 |
| SwinV2-S | gte-base-v1.5 | Freezed | MLP | 94.69 | 89.99 | 89.08 | 89.55 | 92.79 | 91.22 |
| SwinV2-S | gte-base-v1.5 | Freezed | Graph | 95.62 | 90.94 | 88.95 | 89.43 | 92.57 | 91.50 |

For fair comparison to previous works, we use SwinV2-T as the image backbone. We also exemplify how our model scales-up when the bigger, SwinV2-S, image backbone is used. We assess the framework's performance with or without fine-tuning applied to the text backbone. We explore two potential text backbones: gte-base-v1.5 Li et al. (2023) and CLIP ViT-B/32 Radford et al. (2021). The optimal configuration appears to be the frozen gte-base-v1.5 as the text backbone, yielding superior performance. Interestingly, although gte-base-v1.5 boasts approximately 139 million trainable parameters compared to the 63 million parameters in the text module of CLIP ViT-B/32 (totaling 150 million parameters), training with either as a frozen text backbone consumed similar execution times. Memory usage for loading both models required a similar volume. However, fine-tuning both models incurred substantial costs in terms of memory as well as in execution time, without yielding any performance improvements in both text backbones. The drop in performance can possibly be attributed to the fact that the text backbones suffer from overfitting during their tuning. This is somewhat expected as language models usually train on larger datasets over longer training sessions.

We also tested the original MLP transformer decoder architecture as in Shi et al. (2023) with the best performing setting. Memory consumption and execution time using this transformer decoder were comparable to the graph transformer decoder. We find that utilizing the graph structure via the graph transformer decoder as in Hirschorn & Avidan (2023) slightly boosts the performance. Full results are presented in Table 5.

Table 6: **MP-100 results on a bigger image backbone:** $PCK_{0.2}$ performance under the 1-shot setting, using SwinV2-S as the image backbone. Our approach outperforms all other methods on average.

| Model | Support | Split 1 | Split 2 | Split 3 | Split 4 | Split 5 | Avg |
|---|---|---|---|---|---|---|---|
| CapeFormer-S Hirschorn & Avidan (2023) | Image | 92.88 | 89.11 | 89.16 | 87.19 | 88.73 | 89.41 |
| GraphCape-S Hirschorn & Avidan (2023) | Image | 93.66 | 90.42 | **89.79** | 88.68 | 89.61 | 90.43 |
| **CapeX-S** | Text | **95.62** | **90.94** | 88.95 | **89.43** | **92.57** | **91.50** |

In Table 6, we evaluate how our model scales when a larger image backbone is used, comparing it to a visual-prompt-based model with graphs (GraphCape-S) and one without graphs (CapeFormer-

S). Our model maintains its superiority, though the performance gap is smaller in the -S version (+1.07%, +2.09%) compared to the -T version (+1.58%, +2.56%). This is because both GraphCape-S and CapeFormer-S experience performance degradation in both the support and query feature extraction processes, while our model suffers only from a performance drop in the query feature extraction.

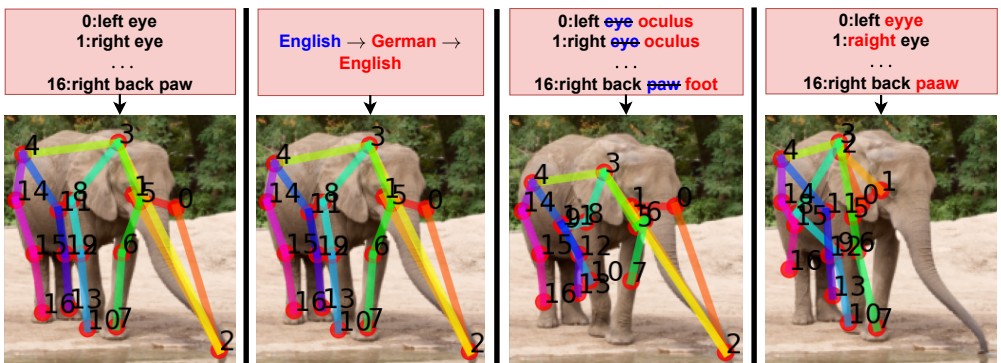

Figure 9: **Modified text descriptions:** Top is the support keypoints text descriptions. Left is a prediction with the **original** text annotation, followed by **translation** test annotations, **synonym** test annotations, and **typo** test annotations, respectively. Below each description, query output(s) are presented.

### A.3.2 Cross Super-Category Pose Estimation

Table 7: **Cross super-category pose estimation results:** $PCK_{0.2}$ performance under the 1-shot setting, on split 1.

| Model | Support | Human Body | Human Face | Vehicle | Furniture |
|---|---|---|---|---|---|
| POMNet Xu et al. (2022) | Image | 73.82 | 79.63 | 34.92 | 47.27 |
| CapeFormer Shi et al. (2023) | Image | 83.44 | 80.96 | 45.40 | 52.49 |
| ESCAPE Nguyen et al. (2024) | Image | 80.60 | 84.13 | 41.39 | 55.49 |
| MetaPoint+ Chen et al. (2024) | Image | 84.32 | 82.21 | 46.51 | 53.67 |
| SDPNet Ren et al. (2024) | Image | 83.84 | 81.24 | 45.53 | 53.08 |
| PPM Peng et al. (2024) | Image (+Text) | 85.13 | 82.44 | 52.08 | 60.59 |
| SCAPE Liang et al. (2025) | Image | 84.24 | 85.98 | 45.61 | 54.13 |
| GraphCape Hirschorn & Avidan (2023) | Image+Graph | 88.38 | 83.28 | 44.06 | 45.56 |
| **CapeX** | Text+Graph | 78.40 | 81.10 | 24.66 | 37.88 |

We build upon the cross super-category evaluation experiments conducted in prior studies to assess the generalization capability of our method across significantly different classes. Specifically, we designate one of the four supercategories in the MP-100 dataset—human face, human body, vehicle, or furniture—as the test set, while the remaining categories are used for training.

Our model was trained with fine-tuning applied to the text backbone over 50 epochs. A comparative analysis with prior methods is presented in Table 7.

This ablation highlights a limitation of our approach: it struggles to resolve symmetry when no "similar" examples are available. For instance, in image-based methods, providing the front left leg of a table as a support keypoint makes it easier to identify the corresponding keypoint, compared to relying on the textual label 'front left leg.' This discrepancy arises because spatial terms like 'left,' 'right,' 'front,' and 'back' can be interpreted inconsistently. Aligning with this observation, we found that fine-tuning the text backbone enhances performance in such scenarios, as it aligns the textual descriptions more effectively with the required keypoints.

### A.3.3 OCCLUSIONS

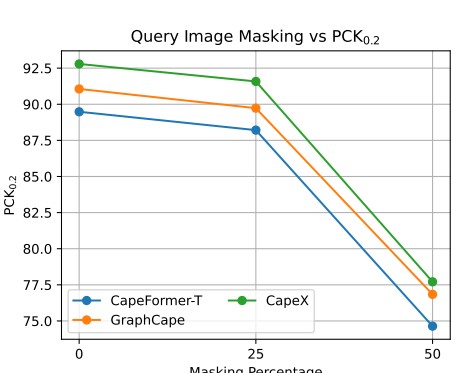

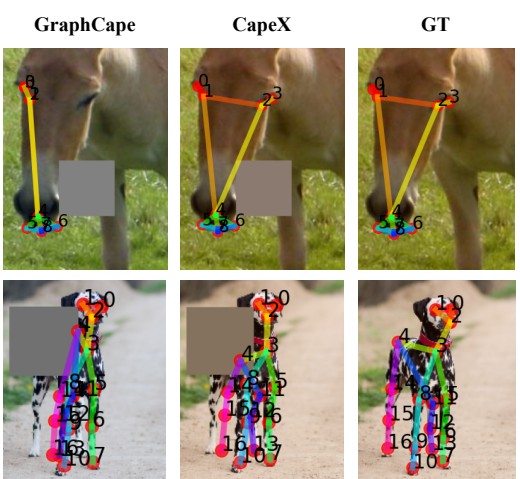

Figure 10: **Masking the query image:** $PCK_{0.2}$ performance as a function of the masking percentage.

Figure 11: **Comparison to GraphCape:** Qualitative comparison between our model and Graph-Cape on query images with 25% masked occlusion. CapeX does not require support images and can handle masked occlusions. Support images and text-graphs are not shown.

We assess the effectiveness of text-graphs in handling occlusions within query images by applying random masks to them before estimating the support keypoints. Quantitative results are in Figure 10, and a qualitative comparison is presented in Figure 11. Our method demonstrates superior performance over GraphCape and CapeFormer-T in the entire presented occlusion range while maintaining similar degradation levels between 0% to 25%. The improved performance at lower masking percentages can be attributed to the text-graphs' abstraction capability and their ability to estimate missing keypoints relative to the visible ones. However, as our approach does not utilize a support image as input, performance significantly drops when a substantial portion of the image is occluded (50%). This is because the absence of the query image leaves the model with insufficient information to operate effectively. This stands in contrast to traditional CAPE methodologies that incorporate a support image, which provides crucial structural cues. In such frameworks, the support image aids the model in hallucinating and extrapolating matching keypoints within the query, particularly when considering graph structures as in GraphCape.

### A.3.4 ADAPTABILITY TO SUPPORT TEXT MODIFICATIONS

We provide additional examples of the ability of our model to adapt to different types of text modifications in Figure 9.

### A.3.5 LEVELS OF ABSTRACTION

We include additional results of our model's performance on different levels of abstractions in Figure 12.

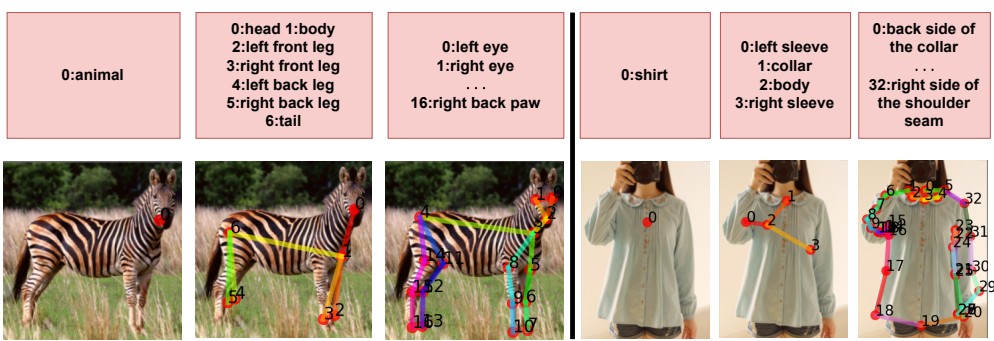

Figure 12: **Text Abstractions:** Model performance over different levels of text-pose abstractions.

