# OpenReview forum: "CapeX: Category-Agnostic Pose Estimation from Textual Point Explanation"
_ICLR.cc/2025/Conference — ICLR 2025 Poster_

### Official Review · Reviewer_J66s · 2024-10-18

**Soundness:** 3
**Presentation:** 3
**Contribution:** 3
**Rating:** 8
**Confidence:** 5

**Summary:**

The paper focuses on the category-agnostic pose estimation task, which learns to estimate the keypoints of novel classes based on a few support images and text descriptions.
Specifically, the paper proposes a pose-graph and use nodes to represent keypoints that are described with text, which could take advantage of the abstraction of text descriptions and the structure imposed by the graph.
On the MP-100 benchmark dataset, the paper achieves a notable performance boost of 1.27% against SOTA methods of category-agnostic pose estimation.

**Strengths:**

- The paper proposes to make full use of text to boost CAPE, which seems novel and interesting.
- The proposed method seems resonable and works well for CAPE.
- The achieved performances are significant against existing works of CAPE.
- The paper is well-written.

**Weaknesses:**

- The description about the related work MetaPoint could be improved, which actually first assigns and then refines.
- The paper composing could be improved. E.g., Fig. 3 and Fig. 4 waste too much space and provide limited information.
- The limitations on using text could be discussed more deeply, because some keypoints are hard to describe using text.
- The experiments could be enhanced with cross super-class transfer as in MetaPoint, CapeFormer and POMNet.

**Questions:**

See Weaknesses*. Overall, the idea of paper seems interesting, and I would like to see more future works. My rating could be increased if my concerns are solved.

---

> ### Author Response · Authors · 2024-11-19
>
> Point 1 and Point 2:
>
> We thank the reviewer for the constructive comments. We will fix these in the next revision.
>
> Point 3:
>
> Most keypoint descriptions in our study were derived from the original MP100 dataset, while others were manually annotated by the authors. A key limitation of using text as support information for arbitrarily annotated keypoints is the semantic similarity among descriptions of similar keypoints. For animal categories, the distinctions between keypoints are generally clear, as demonstrated by the following examples:
>
> ['left eye', 'right eye', 'nose', 'neck', 'root of tail', 'left shoulder', 'left elbow',
> 'left front paw', 'right shoulder', 'right elbow', 'right front paw', 'left hip', 'left knee',
> 'left back paw', 'right hip', 'right knee', 'right back paw']
>
> However, for other categories, such as short_sleeved_shirt, these distinctions can be far less clear, with multiple keypoints sharing similar descriptions, such as right/left sleeve:
>
> ['back side of the collar', 'left side of the collar', 'front left side of the collar',
> 'front side of the collar', 'front right side of the collar', 'right side of the collar',
> 'left side of the shoulder seam', 'left sleeve', 'left sleeve', 'left sleeve',
> 'left sleeve', 'left sleeve', 'left and upper body of the shirt',
> 'left and lower body of the shirt', 'bottom left side of the shirt',
> 'bottom side of the shirt', 'bottom right side of the shirt',
> 'right and lower body of the shirt', 'right and upper body of the shirt',
> 'right sleeve', 'right sleeve', 'right sleeve', 'right sleeve', 'right sleeve',
> 'right side of the shoulder seam']
>
> The complete set of added annotations, including the 68 keypoints for the human face (which also exhibit extensive repetition), is provided in the supplemental material.
>
> While these repetitive annotations may challenge text-based CAPE approaches, our text-graph method addresses this issue by effectively learning relationships among keypoints with similar or identical names. For instance, our approach can relate:
>
> 'left side of the shoulder seam', 'left sleeve', 'left sleeve', 'left sleeve', 'left sleeve', 'left sleeve', 'left and upper body of the shirt',
>
> and appropriately match them to the query image, as demonstrated in Figure 7 (right panel).
>
> Additionally, we evaluate our model using custom skeleton connections. Specifically, testing with an empty graph configuration (representing text-only support) yields a PCK0.2 of 91.27 on split1, compared to 92.79 with the original skeleton. These results underscore the importance of graph structures in effectively managing repetitive annotations and improving performance.
>
>
> Point 4:
>
> Method         | Human Body | Human Face | Vehicle | Furniture
>
> POMNet        | 73.82            | 79.63             | 34.92   | 47.27
>
> CapeFormer  | 83.44            | 80.96             | 45.40   | 52.49
>
> ESCAPE       | 80.60            | 84.13             | 41.39   | 55.49
>
> MetaPoint+    | 84.32            | 82.21             | 46.51   | 53.67
>
> PPM              | 85.13            | 82.44             | 52.08   | 60.59
>
> GraphCape   | 88.38            | 83.28             | 44.06   | 45.56
>
> CapeX           | 78.40            | 81.10             | 24.66   | 37.88
>
>
> This ablation shows a limitation of our approach: our method doesn’t break symmetry correctly when no “similar” examples are present. In image-based approaches, when the front left leg of a table is provided as a support keypoint, it is easier to match the correct keypoint, in comparison to matching based on the text ‘front left leg’. This is because ‘left’/ ‘right’ or ‘front’/ ‘back’ can be interpreted differently. Consistently with this insight, we found that finetuning the text backbone increases performance in this setting since it better fits the textual description to the requested keypoints.

---

> > ### Comment · Reviewer_J66s · 2024-11-24
> > **Reply to the rebuttal**
> >
> > Thanks for the rebuttal. Most of my concerns are addressed, and thus I improve my rating. Besides, I think it is interesting to further solve the repetitive annotation issue for text-based CAPE.

---

> > > ### Author Response · Authors · 2024-11-26
> > >
> > > We sincerely thank the reviewer for their constructive feedback.
> > > We have addressed the concerns regarding the limitations of our work and the advantages of combining text and graphs as support for CAPE.
> > > All revisions have been made in the manuscript and are marked in red for ease of review.

---

### Official Review · Reviewer_dTU9 · 2024-10-30

**Soundness:** 3
**Presentation:** 3
**Contribution:** 3
**Rating:** 6
**Confidence:** 4

**Summary:**

Recent CAPE works have produced object poses based on arbitrary keypoint definitions annotated on a user-provided support image. Our work departs from conventional CAPE methods, which require a support image, by
adopting a text-based approach instead of the support image. Specifically, we use
a pose-graph, where nodes represent keypoints that are described with text. This
representation takes advantage of the abstraction of text descriptions and the structure imposed by the graph.

Our approach effectively breaks symmetry, preserves
structure, and improves occlusion handling. We validate our novel approach using the MP-100 benchmark, a comprehensive dataset covering over 100 categories
and 18,000 images. MP-100 is structured so that the evaluation categories are unseen during training, making it especially suited for CAPE. Under a 1-shot setting,
our solution achieves a notable performance boost of 1.27%, establishing a new
state-of-the-art for CAPE. Additionally, we enhance the dataset by providing text
description annotations for both training and testing. We also include alternative
text annotations specifically for testing the model’s ability to generalize across
different textual descriptions, further increasing its value for future research.1
.

**Strengths:**

1. The motivation is clear  and the paper reads smooth.

2. The design of the presented method is reasonable.

3. Experiments are conducted on the standard MP-100 dataset, establishing a new state-of-the-art for CAPE. The presented method shows superiority to previous CAPE models. Especially, the experimental analysis (e.g. Figure 6 and 7) is very interesting.

**Weaknesses:**

1. The novelty of the presented paper is a little bit concerning. (1) Open-vocabulary (textual prompt) keypoint estimation is not new, there are works such as CLAMP (Zhang et al. 2023) and KDSM (Zhang et al. 2024). (2) Graph representation for few-shot/zero-shot keypoint estimation is not new, as there are works like GraphCape (Hirschorn & Avidan, 2023). Could you highlight the difference between these methods?

2.  Pose-Graph seems to be the most important contribution. However, experimental analysis about the pose-graph is insufficient. For example, there are different graph structures, what if using a fully-connected graph AND different graph structural settings (e.g. Tree-structure)?

3. The reviewer suggests adding "Cross Super-Category Pose Estimation" on MP-100 evaluation, by training the model on all but one super-categories, and evaluate the performance on the remaining one super-category. This is to further evaluate the generalization ability.

4. Since there already exists an open-vocabulary keypoint detection work like CLAMP (Zhang et al. 2023), what is the performance when comparing your method to CLAMP on Animal pose dataset in the setting of five-leave-one-out problem? Namely training on four species while testing on the left one species. The results of PCK @ 0.1 should be presented in paper for comparisons.

[Zhang et al. 2023] CLAMP+Prompt-based Contrastive Learning for Connecting Language and Animal Pose @ CVPR'23

**Questions:**

1. It seems that both visual prompt (support images) and text prompt (keypoint name) have their own limitations. For example, human face may have 68 keypoints, and it seems hard for textual prompts to precisely describe the keypoints along the edge of the face. How to solve this? Could you please give some examples about how to describe these keypoints in language?

2. Figure 6 seems interesting. If I understand correctly,  the experiments about "Modified text descriptions" are evaluated based on the same model without retraining. Is it correct?

3. In "Modified text descriptions", could you add experiments with more detailed description (longer description) about the keypoint. The reviewer is curious about the generalization ability of the system.

---

> ### Author Response · Authors · 2024-11-19
>
> Point 1:
>
> Graphs and texts have been independently explored in previous CAPE studies. In this work, we propose a novel text-graph representation that fuses both modalities. We hypothesize that text-graphs provide an additional layer of abstraction by combining the semantic richness of text with the structural relationships captured by graphs. This enables a more comprehensive representation of semantic keypoints and their interrelations.
> Through an extensive set of experiments, we demonstrate the effectiveness of our approach, achieving state-of-the-art results compared to prior methods relying on support information derived from images, graphs, or text alone. Specifically:
> Image-graph approaches lacked the abstraction and expressiveness that textual information provides (as discussed in Sec. 3.1).
> Text-only approaches faced two primary challenges:
>
> They lacked structural information to represent relationships between keypoints.
>
> They struggled to distinguish between keypoints with similar semantic descriptions.
> For example, text-based methods often fail to adequately differentiate among keypoints like:
>
> ['left side of the shoulder seam', 'left sleeve', 'left sleeve', 'left sleeve', 'left sleeve', 'left sleeve', 'left and upper body of the shirt']
>
> In contrast, our text-graph approach effectively learns relationships between such repetitive keypoints and matches them appropriately to the query image, as illustrated in Figure 7 (right panel). Notably, as discussed in point #2, an empty graph (text-only support) underperforms compared to configurations utilizing meaningful graph connections.
> By addressing these limitations, we believe our work advances CAPE research by presenting a more robust and integrated way to model support input. Our approach sets a new benchmark, paving the way for further innovation in modeling keypoint relationships for diverse annotations.
>
> Point 2:
>
> We understand the reviewer’s concern regarding manual graph annotation. However, graph annotations are required per category, not per instance. In fact, for the whole MP-100 dataset, the authors of GraphCape were required to manually annotate only 22 missing skeleton definitions, since connectivity data was available for most categories.
>
> Furthermore, unlike previous image-based CAPE approaches, our text-graph approach allows for the easy integration of automatically generated graph structures. Since our support input consists of a list of texts representing keypoints (in comparison to locations within an image), we can simply query an LLM model to construct a skeleton suited for the provided texts. We tested how our trained model (on split1) performed on the test set where the connectivity information for each category was replaced with the custom edges that were produced by ChatGPT (provided with only the texts), as well as with a fully connected skeleton, and with an empty graph. The results are:
>
> User Annotated Graphs: 92.79 (original),
>
> LLM-created Skeletons: 91.76,
>
> Empty Graph (no edges): 91.27,
>
> Fully-Connected Graph: 90.23.
>
> This suggests that our model indeed utilizes the graph information while preserving a decent performance with graphs that differ from the training data.
>
>
> Point 3:
>
> We trained our model and finetuned the text backbone, for 50 epochs, and compared it to previous methods:
>
>
> Method         | Human Body | Human Face | Vehicle | Furniture
>
> POMNet        | 73.82            | 79.63             | 34.92   | 47.27
>
> CapeFormer  | 83.44            | 80.96             | 45.40   | 52.49
>
> ESCAPE       | 80.60            | 84.13             | 41.39   | 55.49
>
> MetaPoint+    | 84.32            | 82.21             | 46.51   | 53.67
>
> PPM              | 85.13            | 82.44             | 52.08   | 60.59
>
> GraphCape   | 88.38            | 83.28             | 44.06   | 45.56
>
> CapeX           | 78.40            | 81.10             | 24.66   | 37.88
>
> This ablation shows a limitation of our approach: our method doesn’t break symmetry correctly when no “similar” examples are present. In image-based approaches, when the front left leg of a table is provided as a support keypoint, it is easier to match the correct keypoint, in comparison to matching based on the text ‘front left leg’. This is because ‘left’/ ‘right’ or ‘front’/ ‘back’ can be interpreted differently. Consistently with this insight, we found that finetuning the text backbone increases performance in this setting since it better fits the textual description to the requested keypoints.

---

> > ### Author Response · Authors · 2024-11-19
> >
> > Point 4:
> >
> > We refrained from testing our model on an animal-specific dataset, to keep our focus on a CAPE application rather than a category-specific pose estimation application. However, we tested our model on the animals' dataset, ap10k (which is a subset of mp100), and compared ourselves to CLAMP. We observe lower performance: AP: 47.04, AP .5: 83.36, AP .75: 47.75, AP (M): 28.52, AP (L): 48.36, AR: 58.07 (compared to Table 1 in CLAMP).
> > We attribute CLAMP’s superiority in this setting to the fact that CLAMP is best suited to work on a predefined set of keypoints, which actually makes it the opposite of open-vocabulary. This is due to the fact that it utilizes contrastive loss, thus distinguishing some keypoints from others while training. Indeed, in a CAPE setting that our work is designed for, CLAMP did not perform well, as can be seen in Table 1.
> > In addition, for a fair comparison to the text-based approach KDSM, we tested our model on MP78 in Table 2. Our model outperforms KDSM by a wide margin.

---

> > > ### Comment · Reviewer_dTU9 · 2024-11-24
> > > **Please include these discussions in the main text.**
> > >
> > > Thank you for the response. Glad to see that the authors can honestly and frankly demonstrate the limitations of the proposed method. The reviewer believes these discussions about the limitations are very important and should be included in the main text. I am willing to accept this paper, if the author promises to add these discussions in this main text.

---

> > > > ### Author Response · Authors · 2024-11-26
> > > >
> > > > We sincerely thank the reviewer for their constructive feedback.
> > > > We have addressed the concerns regarding the limitations of our work and the advantages of combining text and graphs as support for CAPE.
> > > > All revisions have been made in the manuscript and are marked in red for ease of review.

---

> > > > ### Author Response · Authors · 2024-12-01
> > > >
> > > > Dear reviewer, we appreciate your willingness to accept our work.
> > > > We have addressed your suggestion and included an extended discussion on the limitations of our approach in our updated submission.
> > > > Please note that the discussion period is ending soon, and we would like to resolve any remaining issues.
> > > > Thank you.

---

### Official Review · Reviewer_5qXx · 2024-11-02

**Soundness:** 3
**Presentation:** 4
**Contribution:** 3
**Rating:** 8
**Confidence:** 5

**Summary:**

This paper introduces a novel approach that diverges from traditional CAPE methods, which typically rely on a support image, by utilizing a text-based framework instead. By implementing a pose-graph in which nodes signify keypoints described through textual annotations, this method leverages the abstraction inherent in text and the structural organization provided by the graph. This strategy effectively mitigates symmetry issues, maintains structural integrity, and enhances the handling of occlusions. The authors enrich the dataset by incorporating text description annotations for both the training and testing phases. They also provide alternative text annotations to specifically evaluate the model's ability to generalize across varying textual descriptions.

**Strengths:**

1. This paper addresses a very important issue: relying on visual information from a support image to locate keypoints in a test image is not reliable.

2. The use of a graph and text representation is intriguing, and I believe it provides a simple yet effective solution to the challenges associated with relying solely on visual information for localization.

3. The paper expands the MP-100 dataset, enabling graph-based CAPE.

4. The experiments in the paper are thorough, and the comparisons are comprehensive.

**Weaknesses:**

I don’t have many questions regarding this paper, just a few minor issues.

1. Were all the keypoint texts in the dataset generated by off-the-shelf foundation models? When many keypoints are very close to each other and have highly similar semantics, how are these very similar keypoints distinguished in the text? Could you provide some examples?

2. The authors could provide more analysis regarding the limitations of the paper, explaining why certain failure cases occur.

**Questions:**

See Weaknesses.

---

> ### Author Response · Authors · 2024-11-19
>
> Point 1:
>
> Most keypoint descriptions in our study were derived from the original MP100 dataset, while others were manually annotated by the authors. A key limitation of using text as support information for arbitrarily annotated keypoints is the semantic similarity among descriptions of similar keypoints. For animal categories, the distinctions between keypoints are generally clear, as demonstrated by the following examples:
>
> ['left eye', 'right eye', 'nose', 'neck', 'root of tail', 'left shoulder', 'left elbow',
> 'left front paw', 'right shoulder', 'right elbow', 'right front paw', 'left hip', 'left knee',
> 'left back paw', 'right hip', 'right knee', 'right back paw']
>
> However, for other categories, such as short_sleeved_shirt, these distinctions can be far less clear, with multiple keypoints sharing similar descriptions, such as right/left sleeve:
>
> ['back side of the collar', 'left side of the collar', 'front left side of the collar',
> 'front side of the collar', 'front right side of the collar', 'right side of the collar',
> 'left side of the shoulder seam', 'left sleeve', 'left sleeve', 'left sleeve',
> 'left sleeve', 'left sleeve', 'left and upper body of the shirt',
> 'left and lower body of the shirt', 'bottom left side of the shirt',
> 'bottom side of the shirt', 'bottom right side of the shirt',
> 'right and lower body of the shirt', 'right and upper body of the shirt',
> 'right sleeve', 'right sleeve', 'right sleeve', 'right sleeve', 'right sleeve',
> 'right side of the shoulder seam']
>
> The complete set of added annotations, including the 68 keypoints for the human face (which also exhibit extensive repetition), is provided in the supplemental material.
>
> While these repetitive annotations may challenge text-based CAPE approaches, our text-graph method addresses this issue by effectively learning relationships among keypoints with similar or identical names. For instance, our approach can relate:
> 'left side of the shoulder seam', 'left sleeve', 'left sleeve', 'left sleeve', 'left sleeve', 'left sleeve', 'left and upper body of the shirt',
>
> and appropriately match them to the query image, as demonstrated in Figure 7 (right panel).
>
> Additionally, we evaluate our model using custom skeleton connections. Specifically, testing with an empty graph configuration (representing text-only support) yields a PCK0.2 of 91.27 on split1, compared to 92.79 with the original skeleton. These results underscore the importance of graph structures in effectively managing repetitive annotations and improving performance.
>
> Point 2:
>
> We observed that our model fails when the images are from vastly different categories from the ones seen during training (Fig. 9). Yet, this is common to all CAPE methods and doesn’t relate to using text-graphs. Fail cases of using text-graphs data emerge when using the wrong descriptions (e.g. in the typo example in Fig. 6) or skeleton data (as was reported in GraphCape).
>
> In addition, we evaluated our model using the cross super-category ablation.
> We trained our model and finetuned the text backbone, for 50 epochs, and compared it to previous methods:
>
>
> Method         | Human Body | Human Face | Vehicle | Furniture
>
> POMNet        | 73.82            | 79.63             | 34.92   | 47.27
>
> CapeFormer  | 83.44            | 80.96             | 45.40   | 52.49
>
> ESCAPE       | 80.60            | 84.13             | 41.39   | 55.49
>
> MetaPoint+    | 84.32            | 82.21             | 46.51   | 53.67
>
> PPM              | 85.13            | 82.44             | 52.08   | 60.59
>
> GraphCape   | 88.38            | 83.28             | 44.06   | 45.56
>
> CapeX           | 78.40            | 81.10             | 24.66   | 37.88
>
>
>
>
>
> This ablation shows a limitation of our approach: our method doesn’t break symmetry correctly when no “similar” examples are present. In image-based approaches, when the front left leg of a table is provided as a support keypoint, it is easier to match the correct keypoint, in comparison to matching based on the text ‘front left leg’. This is because ‘left’/ ‘right’ or ‘front’/ ‘back’ can be interpreted differently. Consistently with this insight, we found that finetuning the text backbone increases performance in this setting since it better fits the textual description to the requested keypoints.

---

> > ### Comment · Reviewer_5qXx · 2024-11-20
> >
> > Oh, I see. The key lies in the graph representation, as it can perceive and distinguish identical names based on the structure and relationships between points. I think this part of the discussion is very important and hope it can be included in the main text later.

---

> > > ### Author Response · Authors · 2024-11-26
> > >
> > > We sincerely thank the reviewer for their constructive feedback.
> > > We have addressed the concerns regarding the limitations of our work and the advantages of combining text and graphs as support for CAPE.
> > > All revisions have been made in the manuscript and are marked in red for ease of review.

---

### Official Review · Reviewer_CrNB · 2024-11-02

**Soundness:** 3
**Presentation:** 3
**Contribution:** 2
**Rating:** 5
**Confidence:** 4

**Summary:**

This paper proposes to employ textual information connected in a graph structure to tackle the problem of category-agnostic pose estimation (CAPE). The textural keypoint descriptions are annotated on the MP-100 dataset. The experimental results on the MP-100 dataset demonstrate the effectiveness of the proposed method.

**Strengths:**

This work focuses on the interesting and important task of category-agnostic pose estimation (CAPE). The proposed CapeX utilizes the abstract textual description for keypoint detection to improve the human-computer interaction. Graph structure is applied to capture the relationship between keypoints. The proposed CapeX achieves the state-of-the-art performance on the MP-100 dataset.

**Weaknesses:**

Pose Anything has designed the graph structure to capture the keypoint correlations, and the textual prompts have been explored for pose estimation in recent works such as KDSM. X-Pose has also provided textual annotations on the MP-100 dataset. Therefore, the contribution of this paper is somewhat incremental.

The graph construction is critical for graph-based approach. Also, manual design of node connections may introduce extra empirical knowledge, leading to unfair comparison. To avoid the above misgiving, the details of graph construction are expected to be clarified.

The cross super-category evaluation on the MP-100 dataset is absent. The cross super-category results would further validate the robustness of the proposed method.

**Questions:**

Please see weaknesses.

---

> ### Author Response · Authors · 2024-11-19
>
> Point 1:
>
> Graphs and texts have been independently explored in previous CAPE studies. In this work, we propose a novel text-graph representation that fuses both modalities. We hypothesize that text-graphs provide an additional layer of abstraction by combining the semantic richness of text with the structural relationships captured by graphs. This enables a more comprehensive representation of semantic keypoints and their interrelations.
> Through an extensive set of experiments, we demonstrate the effectiveness of our approach, achieving state-of-the-art results compared to prior methods relying on support information derived from images, graphs, or text alone. Specifically:
> Image-graph approaches lacked the abstraction and expressiveness that textual information provides (as discussed in Sec. 3.1).
> Text-only approaches faced two primary challenges:
>
> They lacked structural information to represent relationships between keypoints.
>
> They struggled to distinguish between keypoints with similar semantic descriptions.
> For example, text-based methods often fail to adequately differentiate among keypoints like:
>
> ['left side of the shoulder seam', 'left sleeve', 'left sleeve', 'left sleeve', 'left sleeve', 'left sleeve', 'left and upper body of the shirt']
>
> In contrast, our text-graph approach effectively learns relationships between such repetitive keypoints and matches them appropriately to the query image, as illustrated in Figure 7 (right panel). Notably, as discussed in point #2, an empty graph (text-only support) underperforms compared to configurations utilizing meaningful graph connections.
> By addressing these limitations, we believe our work advances CAPE research by presenting a more robust and integrated way to model support input. Our approach sets a new benchmark, paving the way for further innovation in modeling keypoint relationships for diverse annotations.
>
>
> Point 2:
>
> We understand the reviewer’s concern regarding manual graph annotation. However, graph annotations are required per category, not per instance. In fact, for the whole MP-100 dataset, the authors of GraphCape were required to manually annotate only 22 missing skeleton definitions, since connectivity data was available for most categories.
>
> Furthermore, unlike previous image-based CAPE approaches, our text-graph approach allows for the easy integration of automatically generated graph structures. Since our support input consists of a list of texts representing keypoints (in comparison to locations within an image), we can simply query an LLM model to construct a skeleton suited for the provided texts. We tested how our trained model (on split1) performed on the test set where the connectivity information for each category was replaced with the custom edges that were produced by ChatGPT (provided with only the texts), as well as with a fully connected skeleton, and with an empty graph. The results are:
>
> User Annotated Graphs: 92.79 (original),
>
> LLM-created Skeletons: 91.76,
>
> Empty Graph (no edges): 91.27,
>
> Fully-Connected Graph: 90.23.
>
> This suggests that our model indeed utilizes the graph information while preserving a decent performance with graphs that differ from the training data.
>
>
> Point 3:
>
> We trained our model and finetuned the text backbone, for 50 epochs, and compared it to previous methods:
>
> Method         | Human Body | Human Face | Vehicle | Furniture
>
> POMNet        | 73.82            | 79.63             | 34.92   | 47.27
>
> CapeFormer  | 83.44            | 80.96             | 45.40   | 52.49
>
> ESCAPE       | 80.60            | 84.13             | 41.39   | 55.49
>
> MetaPoint+    | 84.32            | 82.21             | 46.51   | 53.67
>
> PPM              | 85.13            | 82.44             | 52.08   | 60.59
>
> GraphCape   | 88.38            | 83.28             | 44.06   | 45.56
>
> CapeX           | 78.40            | 81.10             | 24.66   | 37.88
>
>
> This ablation shows a limitation of our approach: our method doesn’t break symmetry correctly when no “similar” examples are present. In image-based approaches, when the front left leg of a table is provided as a support keypoint, it is easier to match the correct keypoint, in comparison to matching based on the text ‘front left leg’. This is because ‘left’/ ‘right’ or ‘front’/ ‘back’ can be interpreted differently. Consistently with this insight, we found that finetuning the text backbone increases performance in this setting since it better fits the textual description to the requested keypoints.

---

### Author Response · Authors · 2024-11-19

We sincerely appreciate the detailed review and thoughtful feedback—many of which were independently echoed by multiple reviewers. We believe that providing further clarification will enhance the understanding of our research and its contributions. Additionally, in instances where reviewers raised similar points, some of our responses may appear repetitive, as they address overlapping concerns.

---

### Meta-Review · Area_Chair_12eB · 2024-12-20

**Metareview:**

This work proposes to make full use of text to enhance Category-Agnostic Pose Estimation (CAPE), and the method seems to be novel and interesting. The proposed scheme is reasonable and works for this task. Most of the reviewers pointed out that this work is novel and recommended acceptance. There are some weaknesses in the work, e.g., the limitations of using text need to be discussed more deeply,
and the experiments need to be enhanced with cross super-class transfer. Authors have addressed most of the concerns. AC thus recommend acceptance.

**Additional Comments On Reviewer Discussion:**

Authors have addressed most of the concerns raised by reviewers, and these concerns include adding more experiments, and adding more detailed and deeper analysis. One reviewer still has concerns about the limitations in terms of novelty and experiments after rebuttal, but he/she still mentioned that it is also appropriate to accept this paper.

---

### Decision · Program_Chairs · 2025-01-22

Accept (Poster)